# Longitudinal Morphological Changes in the Adenoids and Tonsils in Japanese School Children

**DOI:** 10.3390/jcm10214956

**Published:** 2021-10-26

**Authors:** Hiroshi Yamada, Masaki Sawada, Masaaki Higashino, Susumu Abe, Tarek El-Bialy, Eiji Tanaka

**Affiliations:** 1Yamada Orthodontic Office, Izumiotsu, Osaka 595-0025, Japan; yamada@88-ortho.com (H.Y.); info@88-ortho.com (M.S.); 2Department of Otorhinolaryngology, Head and Neck Surgery, Osaka Medical and Pharmaceutical University, Takatsuki, Osaka 569-8686, Japan; masaaki.higashino@ompu.ac.jp; 3Department of Comprehensive Dentistry, Tokushima University Graduate School of Biomedical Sciences, Tokushima 770-8504, Japan; susumu.abe@tokushima-u.ac.jp; 4Department of Dentistry, University of Alberta, Edmonton, AB T6G 1C9, Canada; telbialy@ualberta.ca; 5Department of Orthodontics and Dentofacial Orthopedics, Tokushima University Graduate School of Biomedical Sciences, Tokushima 770-8504, Japan

**Keywords:** adenoid, tonsil, lymphoid tissue, cephalogram, nasopharynx, oropharynx, upper airway

## Abstract

The adenoid (Ad) and tonsil (Ts), located in the upper airway, play an important role in immunological protection. These lymphoid tissues grow rapidly, reach a peak of growth at the age of 6–8 years, and decrease in their size thereafter. However, little information is available on the longitudinal growth patterns of Ad and Ts in the general population. This study aimed to evaluate the individual growth of Ad and Ts during childhood using lateral cephalograms taken longitudinally from the same individuals at the ages of 8–12 years. Our results showed that the cross-sectional areas of the Ad, nasopharynx (Np), and oropharynx (Op) significantly increased with age while small changes in the size of Ts were present throughout the study period. In addition, the values of Ad/Np and Ts/Op decreased significantly with age in the elementary school. Furthermore, there was a strong and significant correlation between the Ad/Np ratio and upper airway resistance, indicating the narrowest distance in the upper airway. In conclusion, the airway occupation in Np and Op increased with age due to the increase in the sizes of Np and Op but not the decrease in the sizes of Ad and Ts.

## 1. Introduction

Both the adenoids (Ad) and tonsils (Ts) are present at the opening of the upper airway, which serves as an immunological barrier to foreign antigens passing through this area [1]. In the early phase of growth, the immunological activity of the Ad and Ts increases rapidly, reaching approximately 200% growth by late childhood [2]. Subsequently, the growth and development of the Ad and Ts gradually decrease. However, in some individuals, the Ad and Ts remain overgrown even in adults. Furthermore, the mechanisms of overgrowth and subsequent involution of these tissues have not been clearly identified.

Recently, the average proportions of the Ad and Ts sizes in the upper airway area were evaluated at various developmental stages in the Japanese population. There is a significant decrease in the sizes of the Ad and Ts relative to the upper airway between 6 and 20 years of age [1]. The same research group also conducted a longitudinal observational study of the Ad and Ts sizes in Japanese orthodontic patients aged 6–20 years [3]. They provided the mean and standard deviation of the Ad and Ts sizes in five age-based groups, which are helpful for predicting the growth and development of the Ad and Ts during the growth period. However, this research group did not follow-up on the growth of the Ad and Ts. Limited information is available on the longitudinal growth patterns of the Ad and Ts in the general population without orthodontic treatment.

Several studies have shown that enlarged Ad and Ts influence the general growth and development of the body in direct and indirect aspects [1,4]. Overgrown Ad and Ts lead to a developmental disease in the craniofacial region by obstructing the upper airway, including obstructive sleep apnea syndrome (OSAS). OSAS is an important cause of morbidity in both children and adults. Childhood OSAS differs significantly from that of adults in terms of various parameters such as their symptoms, pathogenesis, diagnosis, treatment, and outcome. The estimated prevalence of childhood OSAS ranges from 3 to 10% [5,6]. However, the relationship between the airway and lymphoid tissue morphology (Ad and Ts) and physique (height and weight) in growing children remains unclear.

Hence, the present study was designed to determine the growth patterns of Ad and Ts in the general population and to evaluate the impact of age, sex, physique, and skeletal pattern on the lymphoid tissue growth using lateral cephalograms taken longitudinally for the same individuals at the ages of 8–12 years. We hypothesized that the sizes of Ad and Ts gradually increased during childhood. The aim of this study was to evaluate the individual growth of Ad and Ts in childhood and to identify the factors affecting lymphoid tissue growth.

## 2. Materials and Methods

The sample for this study was chosen from those previously used by Yamada et al. [7]. Permission for this retrospective clinical study was obtained from the Research Ethics Committee of the University of Alberta (Protocol number: 00108824). All studies were conducted in accordance with the relevant guidelines and regulations. The present study provided the necessary estimates for the sample size. The effect size was used for convenient statistical parametric or non-parametric tests. The effect size of the comparison among the three age groups was considered as medium (0.25) [8]. The statistical power (1-β) was calculated using the G*Power software. Power analysis was based on one-way or two-way repeated measurement analysis of variance (r-ANOVA), with a medium effect size of 0.25, a significance level (type I error) of 0.05, and a power level of 0.8. A power analysis was performed after statistical analysis as a post-hoc test.

In the study of Yamada et al. [7], sampling was performed between 1976 and 1981 in a Japanese primary school (Shinjo Primary School, Katsuragi, Nara, Japan). Sixty male and sixty female children aged from 7 years and 2 months to 8 years and 1 month with a mean age of 7 years and 8 months were randomly recruited to take a series of lateral cephalograms. The exclusion criteria were previous maxillofacial trauma or surgery, syndromes, clefts, and orthodontic treatment. Informed consent was obtained from each child and their parents. For each subject, lateral cephalograms were taken every 2 years for 5 years. The first cephalograms were taken at a mean age of 7 years and 8 months (range, 7 years 2 months to 8 years 1 month when the school children were in the 2nd grade in elementary school). Subsequently, the second and final cephalograms were taken at the mean ages of 9 years 8 months and 11 years 8 months when the school children were in the 4th and 6th grades, respectively. All cephalograms were obtained with the teeth in the intercuspal position. Briefly, each subject’s head was fixed with ear rods, and the head was placed in a position such that the Frankfort horizontal plane was parallel to the floor.

Before the measurements, the intra-examiner reliability of cephalometric analysis was examined on 20 randomly selected cephalograms that were traced by the same examiner, and plotted on three arbitrary points (Nasion, Sella, and Pogonion points), twice with a week interval by the same examiner using the intraclass correlation coefficient (ICC). The ICC was 0.984, confirming the reliability of the selected measurements.

Each lateral cephalogram was blindly traced on acetate paper by one examiner (HY). The tracings were computed with a graphic digitizer (Dolphin Imaging, Dolphin Imaging & Management Solutions, Verona, Italy) by another examiner (MS) to obtain the measurements of craniofacial morphology. The accuracy of the tracing was confirmed by two orthodontic professionals joining in this study as collaborators. All investigators were blinded to the general status of the participants. Figure 1 and Figure 2 indicate the definitions of the points and lines used to identify the variables based on previous studies [1,3,9] and illustrate the linear and area measurements as the variables, respectively, as variables.

Statistical analyses were performed using SPSS (version 22.0; SPSS Inc., Chicago, IL, USA) and the statistical package R (version 4.0.2; available as a free download from https://www.r-project.org/, accessed on 25 September 2020). The normality of the distribution of each cephalometric variable was assessed using the Shapiro–Wilk test and confirmed by an evaluation of the Q–Q plot. If the normality of the data distribution was verified, the data were expressed as the mean and a standard deviation (SD). However, if the data did not follow the normal distribution, data were expressed as the median and interquartile range (IQR: first quartile-third quartile). The differences in body height and weight between boys and girls were evaluated for each grade. For the normal distributed data, a general linear model analysis for repeated measures was performed to compare the three grades. Intergroup comparisons were carried out using the paired *t*-test or pairwise Wilcoxon rank sum test with the Bonferroni method as a post hoc test. Pearson’s correlations of each measurement value with those of the other variables were calculated to assess the correlation strength between the two variables chosen from any cephalometric variables. Probabilities below 0.05, as the type I error (α), were considered statistically significant.

## 3. Results

### 3.1. Participants

Of the 120 school children’s records, this study sample consisted of 49 boys and 50 girls that had lateral cephalometric radiographs taken annually throughout the 5-year study period, and the total sample size used in the present study was 99. In total, 21 children were excluded because their records did not have the height and weight data. To confirm the validity of the sample size in the present study, a power analysis was performed. As a result, the total estimated sample size was determined to be 44 when the one-way r-ANOVA was used. This indicates that the sample size in this study was appropriate, and that the effect size of the present results was large.

For all participants, there were no significant interactions in the values of body height, weight, and body mass index (BMI) between boys and girls. However, body height and weight significantly (*p* < 0.001) increased with age, and the average height and weight did not differ significantly between girls and boys throughout the study period. The BMI was significantly (*p* < 0.01) smaller in girls than in boys, and both values increased significantly (*p* < 0.001) with age (Table 1). In brief, since sex differences were not recognized in the present study, all cephalometric variables were combined with the data of boys and girls.

### 3.2. Cephalometric Measurements of the Lymphoid Tissues and Upper Airway and Their Correlations

For Ts, small changes in size were present throughout the study period, while the size of Ad significantly (*p* < 0.001) increased with age. The average sizes of Ts were 125.91 (105.31–172.75) mm^2^ at the 2nd grade, 134.96 (102.25–177.50) mm^2^ at the 4th grade, and 135.56 (108.17–179.34) mm^2^ at the 6th grade in elementary school, while the average sizes of Ad were 340.16 (293.89–396.83) mm^2^, 333.77 (284.41–398.78) mm^2^, and 386.66 (334.27–467.43) mm^2^ at the 2nd, 4th, and 6th grades, respectively. The cross-sectional areas of Np and Op significantly (*p* < 0.001) increased with age throughout the study period. The cross-sectional area of the nasopharynx (Np) was 1.2 times larger in the 6th grade compared to that in the 2nd grade, while the cross-sectional area of the oropharynx (Op) was 1.2 times larger in the 6th grade in elementary school. The value of Ad/Np decreased significantly (*p* < 0.01) from the 2nd to 4th grade, and increased significantly (*p* < 0.01) again from the 4th to 6th grade in elementary school. The Ts/Op increased significantly (*p* < 0.01) with age.

The 4th and 6th grade children showed significantly larger values of the anteroposterior lengths of the nasopharyngeal airway space (PNS-UPW), upper airway resistance (UAR), and angle of ANB (A-point-Nasion-B-point) than the 2nd grade children, while there were no significant differences in the anteroposterior length of the retropalatal oropharyngeal airway space (U-MPW). Furthermore, the anteroposterior length of the hypopharyngeal airway space (V-LPW) was significantly larger in the 6th grade than in the 2nd and 4th grades. The soft palate thickness (SPT) in the 6th grade was significantly (*p* < 0.05) greater than that in the 2nd grade. However, the U-MPW values did not differ significantly for each grade (Table 2).

### 3.3. Correlations of Continuous Variables

The correlation coefficients were calculated for 12 cephalometric variables and three physical variables using Pearson’s correlation. Among the 105 correlations, 62 were statistically significant (Table 3). In particular, 7 of 62 correlations showed strong and significant (*p* < 0.0001) positive or negative correlations, including Np and Ad, Op and V-LPW, Ts and Ts/Op, body height and weight, body weight and BMI, and Ad/Np and UAR. Four of 62 showed significant (*p* < 0.001) positive or negative correlations with U-MPW and V-LPW, UAR and PNS-UPW, Ad/Np and Ad, and Ad/Np and PNS-UPW. Moreover, 15 of 62 showed weak but significant (*p* < 0.01) correlations among the continuous variables (Table 3).

## 4. Discussion

The evaluation modalities for upper airway and lymphoid tissue morphology include three-dimensional computed tomography (CT), cone-beam CT, and magnetic resonance imaging (MRI) other than cephalometric radiographs [10,11,12,13]. Although accurate 3-dimensional images are obtained when using 3-dimensional CT and cone-beam CT, these techniques have disadvantages such as high radiation exposure and high cost. Considering the high radiation dose, it may be difficult to perform CT repeatedly for each subject without clinical demand. A study that compared cephalometric radiography to MRI suggested that cephalograms have the advantage of avoiding misevaluation of the Op due to overlapping structures [13]. The cephalogram is a simple, expensive, and sufficiently informative diagnostic technique with a low radiation dose, and the generated two-dimensional images are sufficiently reliable and may be an alternative to three-dimensional images for the evaluation of lymphoid tissues and upper airway morphology [12,14]. Savoldi et al. [9] examined the reliability of lateral cephalometric radiographs to assess upper airway and lymphoid tissues and reported a higher reliability of lateral cephalometric radiographs for the measurement of upper airway morphology. These results indicate that cephalograms allow us to perform high-quality evaluations of the upper airway and lymphoid tissue morphology.

In our study, the average sizes of Ad and Ts and the lymphoid tissues–airway ratios were longitudinally analyzed at three ages. The cross-sectional areas of Ts did not change significantly during growth, and those of Ad increased significantly with the grade; however, the lymphoid tissues–airway ratios, including Ad/Np and Ts/Op, decreased significantly between 8-year-olds and 12-year-olds. This implies that the cross-sectional areas of Op and Np increased significantly during growth in our study sample, resulting in an increase in airway occupation in both Np and Op. For the first time, Manabe et al. [1] determined the average proportions of the Ad and Ts sizes to the upper airway area at various developmental stages in Japanese individuals in a cross-sectional study, and demonstrated that the values for the Ad/Np were 60.7 ± 7.8% and 53.2 ± 12.5% for the lower and higher primary school groups, respectively, with a significant difference between the two values. For the Ts/Op values, they also reported that the mean values for the Ts/Op were 32.5 ± 15.1% and 28.0 ± 16.9% for the lower and higher primary school groups, respectively, with no significant difference between them. Our findings are consistent with previously published data on the increased airway occupation of Np and Op areas in school children in terms of the increased airway occupation of Np and Op areas [1]. However, the lymphoid tissues–airway ratio, especially the Ts/Op, differed greatly from the previous study [1]. This is due to the differences in the definitions of the oropharyngeal area. We found cephalograms, in which the tongue had no contact with the soft palate and tonsil. In these cephalograms, it was impossible to define the anterior border of the air area, and we used the line connecting the base point of the tonsil and the vallecular as the anterior border of the air area. Furthermore, the position of the superior point of the epiglottis is very sensitive to the swallowing stage; thus far, we have chosen the vallecula to define the inferior border of the air area.

Ishida et al. [3] showed that the mean values of Ad and Ts were 347.55 ± 12.52 mm^2^ and 161.34 ± 9.54 mm^2^ in the younger primary school group and 346.22 ± 12.63 mm^2^ and 152.82 ± 9.35 mm^2^ in the older primary school group, respectively. Our results showed that the cross-sectional area of Ts was, on average, 125.91 mm^2^ in the 2nd grade, 134.96 mm^2^ in the 4th grade, and 135.56 mm^2^ in the 6th grade. Furthermore, the average cross-sectional area of Ad was 340.16 mm^2^, 333.77 mm^2^, and 386.66 mm^2^ at the 2nd, 4th, and 6th grades, respectively. This means that the Ad and Ts sizes did not decrease significantly between the younger and older primary school groups as well as in a previous study [3]. Therefore, our hypothesis that the sizes of Ad and Ts gradually increases during childhood was not confirmed. Since the present study aimed to determine the growth patterns of Ad and Ts in childhood, an important finding was that the overgrowth of Ad and Ts often appeared in children of younger grades in elementary school. There were no or fewer changes in the size of Ad and Ts at 8–12 years old. This indicates that the growth and developmental patterns of lymphoid tissues are quite different for each individual.

Matsumoto et al. [15] investigated the characteristics of craniofacial architecture in children with OSAS, and demonstrated that 47% of the children with apnea–hypopnea indexes (AHI) ≥ 3 were considered to have skeletal Class II, and that 63% of the children with AHI < 3 were skeletal Class III. Hoekema et al. [16] also indicated that a larger ANB angle was found in children with OSAS. Meanwhile, Lopatiene et al. [17] evaluated the relationships between hard and soft tissues and upper airway morphology in patients with normal sagittal occlusion or Angle Class II malocclusion without OSAS using lateral cephalograms and demonstrated that Angle Class II patients with a significant lip protrusion and decreased facial convexity angle were subjected to airway restriction. Our results showed a weak but significant negative correlation between the UAR and ANB angle in school children. Although the children in our study had no or fewer symptoms of OSAS, school children with skeletal Class II are likely to have a smaller UAR, which means a narrower upper airway, leading to nasal breath obstruction. Furthermore, among the 99 school children, 12 children showed > 70% of the Ad/Np ratio throughout the experimental period (5 years) (Appendix A). Interestingly, 11 of these 12 children also showed < 7.0 mm of the UAR even in the 6th grade, while the mean value of the UAR in the 6th grade was 10.7 ± 3.6 mm. Furthermore, 7 of 12 children exhibited an ANB angle > 5.5°, indicating that they were considered to have a skeletal Class II jaw–base relationship. Based on these considerations, school children with > 70% of the Ad/Np ratio are likely to show a narrower upper airway with skeletal Class II, which may lead to OSAS in childhood. OSAS in children can be managed by the removal of enlarged Ts and Ad, namely adenotonsillectomy [18]. Moreover, a prospective longitudinal clinical study indicated that adenotonsillectomy can improve the facial growth of children with obstructive hypertrophy of lymphoid tissues [19]. Since it is unclear whether this surgical intervention is required for such cases, our results may provide available information about the Ad/Np threshold value for an indication of adenotonsillectomy. Further studies with a larger number of subjects are needed to verify the study result.

## 5. Conclusions

In conclusion, the present study identifies the growth pattern of the Ad and Ts during childhood using lateral cephalograms taken longitudinally at the ages of 8–12 years. During childhood, the cross-sectional areas of the Ad, Np, and Op significantly increased with age, while small changes in the sizes of the Ts were present during childhood. However, airway occupation in the Np and Op significantly increased with age in elementary school. This may have future implications regarding the perioperative assessment of the Ad and Ts and future surgical therapeutic approaches to OSAS in children.

## Figures and Tables

**Figure 1 jcm-10-04956-f001:**
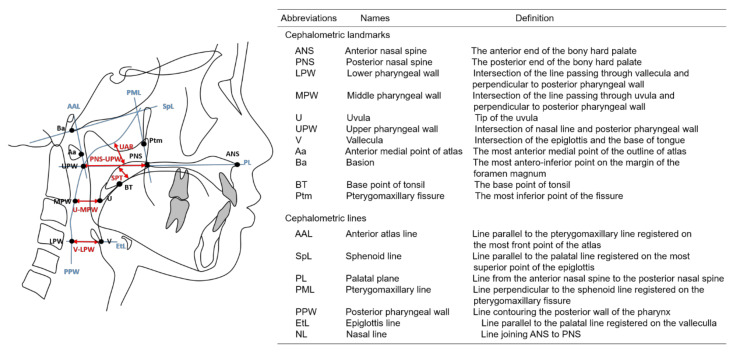
Cephalometric landmarks (black dots), reference lines (blue solid), and linear measurements (red arrows) of the nasopharynx and oropharynx.

**Figure 2 jcm-10-04956-f002:**
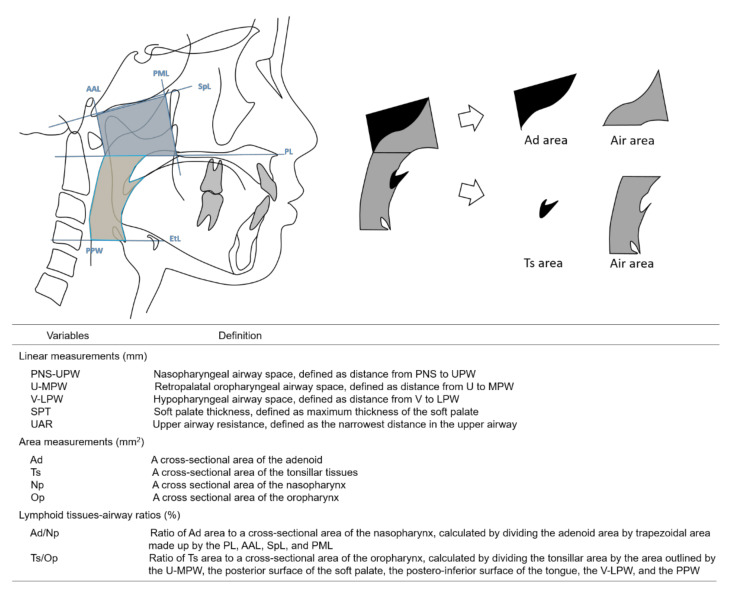
Cephalometric reference lines (blue solid) and area measurements (black solid) and the definitions of the adenoid area to the nasopharyngeal area ratio and the tonsillar area to the oropharyngeal area ratio. If the tongue has no contact with the soft palate and tonsil, the anterior border of the air area is the line connecting between the base point of the tonsil and the vallecula.

**Table 1 jcm-10-04956-t001:** Mean and median values of body measurements and their associations with sex and grade.

		Grade 2	Grade 4	Grade 6	Interaction	Sex	Grade
Height (cm)	Boy	115.14 ± 4.75	120.55 ± 4.95	126.37 ± 5.29	0.654	0.614	<0.001
	Girl	115.69 ± 4.71	121.14 ± 4.97	126.73 ± 5.28			
Weight (cm)	Boy	20.20 (18.25–21.90)	22.40 (20.20–24.50)	25.00 (22.70–27.70)	0.539	0.282	<0.001
	Girl	19.55 (18.38–20.85)	21.60 (20.16–22.85)	24.10 (22.75–25.40)			
BMI (cm)	Boy	15.09 (14.39–15.95)	15.10 (14.57–16.16)	15.68 (14.09–16.77)	0.163	0.007	<0.001
	Girl	14.69 (14.22–15.38)	14.53 (14.13–15.34)	15.00 (14.48–15.59)			

**Table 2 jcm-10-04956-t002:** Mean and median values of cephalometric and body measurements and their relation to grade.

	Grade 2	Grade 4	Grade 6	*p*-Value	Grade 2 vs. 4 ^§^	Grade 4 vs. 6 ^§^	Grade 2 vs. 4 ^§^
PNS-UPW (mm)	20.73 ± 4.17	22.88 ± 4.58	23.28 ± 4.80	<0.001	<0.001	0.828	<0.001
U-MPW (mm)	11.19 ± 3.11	11.26 ± 3.80	11.75 ± 3.46	0.198	<0.001	0.447	0.293
V-LPW (mm)	13.74 (11.38–15.69)	14.31 (12.66–16.02)	15.88 (13.39–18.51)	<0.001	0.767	0.001	0.001
SPT (mm)	7.84 ± 1.12	8.07 ± 1.08	8.24 ± 1.22	0.022	0.290	0.624	0.039
UAR (mm)	8.74 ± 2.89	10.56 ± 3.12	10.68 ± 3.70	<0.001	<0.001	1.000	<0.001
Ad (mm^2^)	340.16 (293.89–396.83)	333.77 (284.41–398.78)	386.66 (334.27–467.43)	<0.001	0.767	0.001	0.001
Ts (mm^2^)	125.91 (105.31–172.75)	134.96 (102.25–177.50)	135.56 (108.17–179.34)	0.399	0.900	1.000	0.714
Nasopharynx (mm^2^)	529.76 (480.20–596.26)	580.50 (523.27–647.47)	656.47 (592.98–735.16)	<0.001	<0.001	<0.001	<0.001
Oropharynx (mm^2^)	524.58 (445.48–644.24)	580.49 (508.00–692.72)	674.03 (575.50–788.62)	<0.001	0.140	0.001	0.001
ANB (^o^)	5.18 ± 2.56	4.73 ± 2.55	4.74 ± 2.59	0.001	0.001	1.000	0.009
Ad/Np (%)	63.34 (58.57–71.53)	56.99 (49.54–65.29)	59.99 (52.67–67.57)	<0.001	0.001	0.032	0.001
Ts/Op (%)	24.41 (18.46–30.16)	23.88 (17.78–29.95)	19.76 (15.56–23.97)	<0.001	0.165	0.001	0.001

Data are expressed as mean ± SD when the data distribution was normal. Generalized linear model analyses for repeated measures are performed to assess the effect of grade. Data are expressed as median (first quartile–third quartile) when the data distribution was out of normal. Brunner-Lunge tests are performed to assess the effect of grade. Mean ± SD shown for normal data distribution. Generalized Linear Mode for repeated measures followed by pairwise mean comparisons were used. Median (first quartile–third quartile) shown when data distribution was not normal. Friedman and Wilcoxon signed rank tests were used. ^§^; Bonferroni corrected *P* values.

**Table 3 jcm-10-04956-t003:** Correlation coefficients between cephalometric and physique body measurements.

Correlation Coefficient	U-MPW (mm)	*V-LPW* (mm)	SPT (mm)	UAR (mm)	*Ad* (mm^2^)	*Ts* (mm^2^)	*Nasopharynx* (mm^2^)	*Oropharynx* (mm^2^)	ANB (^o^)	Height (cm)	*Weight* (kg)	*BMI* (kg/m^2^)	*Ad/Np* (%)	*Ts/Op* (%)
PNS-UPW (mm)	0.248 ***	0.284 ****	−0.087	0.584 ****	−0.130*	0.053	0.416 ****	0.333 ****	0.061	0.221 ***	0.123 *	−0.055	−0.669 ****	−0.201 ***
U-MPW (mm)		0.529 ****	−0.113	0.193 **	0.272 ****	0.460 ****	0.324 ****	0.742 ****	−0.071	0.080	0.089	0.067	0.020	0.007
*V-LPW* (mm)			−0.081	0.157 **	0.168 **	0.218 ***	0.290 ****	0.759 ****	−0.063	0.267 ****	0.229 ****	0.079	−0.092	−0.264 ****
SPT (mm)				−0.184 **	0.136 *	0.051	0.058	−0.083	0.009	0.104	0.130 *	0.112	0.107	0.117 *
UAR (mm)					−0.396 ****	0.032	0.161 **	0.294 ****	−0.153 **	0.220 ***	0.106	−0.085	−0.778 ****	−0.171 **
*Ad* (mm^2^)						0.301 ****	0.742 ****	0.248 ****	0.053	0.329 ****	0.368 ****	0.252 ****	0.605 ****	0.148 *
*Ts* (mm^2^)							0.252 ****	0.495 ****	0.046	0.099	00.110	0.074	0.159 **	0.755 ****
*Nasopharynx* (mm^2^)								0.358 ****	−0.007	0.483 ****	0.437 ****	0.186 **	−0.069	0.015
*Oropharynx* (mm^2^)									−0.090	0.391 ****	0.358 ****	0.162 **	−0.063	−0.137 *
ANB (^o^)										−0.085	−0.087	−0.046	0.094	0.108
Height (cm)											0.882 ****	0.351 ****	−0.106	−0.188 **
*Weight* (kg)												0.745 ****	−0.002	−0.158 **
*BMI* (kg/m^2^)													0.123	−0.063
*Ad/Np* (%)														0.222 ***

Roman style shows the correlation coefficient analyzed by Pearson’s correlation test. * *p* < 0.05; ** *p* < 0.01; *** *p* < 0.001; **** *p* < 0.0001. Italic style shows the correlation coefficient analyzed by Spearman’s correlation test.

## Data Availability

Not applicable.

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
