# Peer review of "Longitudinal Morphological Changes in the Adenoids and Tonsils in Japanese School Children"

_jcm, 2021, doi:10.3390/jcm10214956_

Round 1

Reviewer 1 Report

  • Positive:

       - very interesting topic and approach to the issue,

       - good statistics.

       - connection the results of morphometric tests of the head with height              and weight.

       - morphometric head tests should be published in a modern anatomical            journal.

  • Negativs:     

- the qualification of children for a radiological examination raises enormous ethical questions. Was the only indication for multi radiological examinations given to research or was the examination carried out without medical indications ???????

-no correlation with the clinical picture. The indication for surgery is not only the size of the tonsils or adenoid, but the clinical symptoms (otitis media secretora, frequent infections etc.).

- tonsils are three-dimensional structures. Failure to assess in all dimensions disqualifies from drawing conclusions. The size of the tonsils in mm2 not in mm?????

- the study assessed the dimensions of the nasopharynx and middle pharynx. On this basis, it is difficult to infer the size of the tonsils.

-lack of registration all radiological measurements in the appendix.

- page 7/10, verse208 ??????

- lack of current references.

Author Response

  • Positive:

- very interesting topic and approach to the issue,

- good statistics.

- connection the results of morphometric tests of the head with height and weight.

- morphometric head tests should be published in a modern anatomical journal.

(Response)

Thank you for the nice wordings. However, the present study is cephalometric analysis study but not morphometric head tests.

  • To be addressed:

- the qualification of children for a radiological examination raises enormous ethical questions. Was the only indication for multi radiological examinations given to research or was the examination carried out without medical indications ???????

(Response)

One of the authors worked as school doctor in this primary school, and was in charge of the medical and dental checkup every year. The radiological examinations were done as part of the annual medical records.

-no correlation with the clinical picture. The indication for surgery is not only the size of the tonsils or adenoid, but the clinical symptoms (otitis media secretora, frequent infections etc.).

(Response)

In the otopharyngeal field, the results obtained in this study may not involve the clinical indication for surgery. However, in the orthodontic field, the information about the size of tonsils and adenoids is of clinical importance for the prognosis of orthodontic treatment.

- tonsils are three-dimensional structures. Failure to assess in all dimensions disqualifies from drawing conclusions. The size of the tonsils in mm2 not in mm?????

(Response)

We agree with the reviewer’s comments that tonsils are three-dimensional structure; however, the availability of three-dimensional x-rays is almost impossible, in addition, previous studies showed that lateral cephalometric radiographs, that are 2-dimensional simple radiograph with a low-level radiation exposure compared to a computed tomography, have been used before in the literature to estimate tonsils size. It is true that using computed tomography, we can obtain detailed information on the 3-dimensional size and shape of tonsils and adenoids, while high-dose radiation has to be exposed.

- the study assessed the dimensions of the nasopharynx and middle pharynx. On this basis, it is difficult to infer the size of the tonsils.

(Response)

We thank the reviewer for the valued comment and we agree with the reviewer’s comment. As described in the Discussion section, it is well known that the cephalogram is a simple, expensive, and sufficiently informative diagnostic technique with less radiation dose, and the generated 2-dimensional images are sufficiently reliable and may be an alternative to 3-dimensional images in the evaluation of lymphoid tissues and upper airway morphology. Many researchers examined the reliability of lateral cephalometric radiographs to assess the upper airway and lymphoid tissues, and concluded a higher reliability of cephalometric analysis for the upper airway morphology. These indicate that cephalograms allow us to perform a high-quality evaluation of the upper airway and lymphoid tissue morphology.

-lack of registration all radiological measurements in the appendix.

(Response)

In the appendix (supplemental figure), we added one more graph in which grade-dependent change in the ANB angle was exhibited. (revision: Supplemental Figure 1)

- page 7/10, verse208 ??????

(Response)

We rewrote this sentence into the following:

Savoldi et al. [9] examined the reliability of lateral cephalometric radiographs to assess upper airway and lymphoid tissues and reported a higher reliability of lateral cephalometric radiographs for the measurement of upper airway morphology. (revision: Page 6, lines 199-204)

- lack of current references.

(Response)

We replaced some old references into some current references in the revised version. (revisions: References #5, 6, and 11)

Reviewer 2 Report

This study aimed to examine longitudinal changes in morphology of adenoids and tonsils, also relative to changes in the areas of the nasopharynx and oropharynx. The study used previously collected data (cephalograms) on Japanese School children who were measured multiple times across the period of approximately 5 years. The aim of the study was to examine longitudinal changes in these tissues and to compare these results to previous studies to get a better understanding of general changes in adenoids and tonsils in the general population. The study found an increase in size of adenoids, nasopharynx and oropharynx between 8-12 years. The study adds some additional data on changes in these tissues in a population that was not sampled due to some other reason (e.g. orthodontic treatment)

General queries/comments:
- The paper needs a good line edit. I started to point out grammatical errors (see some examples below) at the beginning of the paper, but realised these occurred throughout the manuscript and would take too long to describe. Because of the grammatical errors, I had to read some sentences several times to understand what the authors were trying to describe. 
A few examples…
Line 18 - change “in general” to “in the general”
Line 22 - change “On the other, the values” to “On the other hand, the ratios”?
Line 24 - change “strongly and significantly” to “strong and significant”

Mixing of statistical tests:
In Table 3 and 4, both means and medians are presented. In Table 5, spearman and Pearson correlation coefficients are presented. Presenting both means and medians is probably ok (justifiable), but I typically don’t see both Spearman/Pearson correlations presented in the same table, usually just one is presented. This might depend on the presentation rules of the journal. 

Figure 1/2 and Table 1/2 - to be more useful, abbreviations in the table need to be combined with (next to) the figures in order to cross reference abbreviations more easily. 

Author Response

This study aimed to examine longitudinal changes in morphology of adenoids and tonsils, also relative to changes in the areas of the nasopharynx and oropharynx. The study used previously collected data (cephalograms) on Japanese School children who were measured multiple times across the period of approximately 5 years. The aim of the study was to examine longitudinal changes in these tissues and to compare these results to previous studies to get a better understanding of general changes in adenoids and tonsils in the general population. The study found an increase in size of adenoids, nasopharynx and oropharynx between 8-12 years. The study adds some additional data on changes in these tissues in a population that was not sampled due to some other reason (e.g. orthodontic treatment)

(Response)

Thank you for the nice wordings.

General queries/comments:

- The paper needs a good line edit. I started to point out grammatical errors (see some examples below) at the beginning of the paper, but realised these occurred throughout the manuscript and would take too long to describe. Because of the grammatical errors, I had to read some sentences several times to understand what the authors were trying to describe. 

A few examples…

Line 18 - change “in general” to “in the general”

Line 22 - change “On the other, the values” to “On the other hand, the ratios”?

Line 24 - change “strongly and significantly” to “strong and significant”

(Response)

According to your advice, we asked English experts (Editage) to revise the manuscript thoroughly. (numerous revisions)

Mixing of statistical tests:

In Table 3 and 4, both means and medians are presented. In Table 5, spearman and Pearson correlation coefficients are presented. Presenting both means and medians is probably ok (justifiable), but I typically don’t see both Spearman/Pearson correlations presented in the same table, usually just one is presented. This might depend on the presentation rules of the journal. 

(Response)

Thank you for your good suggestion. Reviewer is correct. Since this study was managed a lot of parameters, Spearman or Pearson correlations were adapted by whether each parameter was normality data or not. Then, we reanalyzed the relationship between two parameters (cephalometric and physique body measurements) using a graph regardless of the parametric or non-parametric distribution. Pearson correlation coefficient was adopted in this study so that some numbers of the correlation were changed. However, the changes in the results did not affect the discussion. (revisions: New Table 3; Page 6, lines 178-186)

Figure 1/2 and Table 1/2 - to be more useful, abbreviations in the table need to be combined with (next to) the figures in order to cross reference abbreviations more easily. 

(Response)

Thank you for the nice advice. According to your advice, we combined Figure 1 and Table 1 with New Figure 1 and Figure 2 and Table 2 with New Figure 2.

Round 2

Reviewer 1 Report

The authors' comments are acceptable